# Performance of Universal Reciprocating Heat-Engine Cycle with Variable Specific Heats Ratio of Working Fluid

**DOI:** 10.3390/e22040397

**Published:** 2020-03-31

**Authors:** Lingen Chen, Yanlin Ge, Chang Liu, Huijun Feng, Giulio Lorenzini

**Affiliations:** 1Institute of Thermal Science and Power Engineering, Wuhan Institute of Technology, Wuhan 430205, China; geyali9@hotmail.com (Y.G.); lc198707181@126.com (C.L.); huijunfeng@139.com (H.F.); 2School of Mechanical & Electrical Engineering, Wuhan Institute of Technology, Wuhan 430205, China; 3Dipartimento di Ingegneria e Architettura, Universita’ di Parma, Parco Area delle Scienze 181/A, 43124 Parma, Italy; giulio.lorenzini@unipr.it

**Keywords:** finite time thermodynamics, reciprocating heat-engine cycle, universal cycle, variable specific heat ratio, power output, thermal efficiency

## Abstract

Considering the finite time characteristic, heat transfer loss, friction loss and internal irreversibility loss, an air standard reciprocating heat-engine cycle model is founded by using finite time thermodynamics. The cycle model, which consists of two endothermic processes, two exothermic processes and two adiabatic processes, is well generalized. The performance parameters, including the power output and efficiency (PAE), are obtained. The PAE versus compression ratio relations are obtained by numerical computation. The impacts of variable specific heats ratio (SHR) of working fluid (WF) on universal cycle performances are analyzed and various special cycles are also discussed. The results include the PAE performance characteristics of various special cycles (including Miller, Dual, Atkinson, Brayton, Diesel and Otto cycles) when the SHR of WF is constant and variable (including the SHR varied with linear function (LF) and nonlinear function (NLF) of WF temperature). The maximum power outputs and the corresponding optimal compression ratios, as well as the maximum efficiencies and the corresponding optimal compression ratios for various special cycles with three SHR models are compared.

## 1. Introduction

Using finite time thermodynamics (FTT) [1,2,3,4,5,6,7,8,9,10,11,12,13,14,15,16] to optimize the performances of practical cycles and processes, a series of achievements were made, including Novikov heat engines [17,18,19,20,21], Curzon–Ahlborn heat engines [22,23,24], solar-driven engines [25,26], Maisotaenko cycle [27,28,29], OTEC systems [30,31,32], Kalina cycle [33], thermoelectric devices [34,35,36,37,38,39], dissipative heat engine [40], refrigeration cycle [41], earth [42], quantum systems [43,44,45,46,47,48,49,50], economic systems [51,52], chemical systems [53,54,55,56,57,58,59,60,61], reciprocating internal combustion engines [62,63,64,65,66], etc. In the early studies, for the reciprocating heat-engine cycle (RHEC), the specific heats (SH) of working fluid (WF) were usually assumed to be constant. For the practical cycle, the properties and composition of the WF will change with the occurrence of the combustion reaction. So the SH of WF will also change with the occurrence of the combustion reaction, and this change has a great influence on cycle performance. The variation of SH of WF would inevitably cause variation of the performance, so the studies on air standard (AS) RHEC performance analysis and optimization can be divided into three classes according to the SH of WF models, including the constant SH model, variable SH model and variable SHR model, see the book [64] and review article [65] in detail.

In the first class, Klein [67] studied the net power versus efficiency relations of endoreversible Diesel and Otto cycles, and obtained the maximum work output and the corresponding compression ratio (CR). References [68,69] derived the power output and efficiency (PAE) and performance characteristics (PC) of Diesel [68] and Otto [69] cycles with heat transfer loss (HTL). Angulo-Brown et al. [70] modeled the Otto cycle with friction loss (FL), and studied the impact of FL on the cycle performance. Using expansion and compression efficiencies to define the internal irreversibility loss (IIL), Chen et al. [71] studied the irreversible Otto cycle performance. Qin et al. [72] modeled a universal RHEC (including Otto, Diesel, Brayton, and Atkinson cycles) which was consisted of an endothermic process, an exothermic process and two adiabatic processes, and derived the PAE relation with FL and HTL. Reference [73] founded a more universal RHEC (including Miller, Dual, Atkinson, Brayton, Diesel and Otto cycles) which consisted of two endothermic processes, two exothermic processes and two adiabatic processes, derived the PAE and PC, and gave out the maximum power output and the maximum efficiency orders of each special cycle.

In the second class, Ghatak and Chakraborty [74] studied the impact of variable SH of WF with the LF of temperature on the PAE and PC of endoreversible Dual cycle. Considering the variable SH with the LF of temperature, references [75,76] studied the performances of Dual [75] and Miller [76] cycles with FL and HTL. References [77,78,79,80] analyzed the performances of irreversible Otto [77], Atkinson [78], Diesel [79] and Brayton [80] cycles with HTL, IIL and the variable SH of WF with the LF of temperature. Chen et al. [81] established a universal cycle which consisted of two endothermic processes, two exothermic processes and two adiabatic processes with HTL, FL and variable SH of WF and compared the differences of each special cycle performance when the SH of WF were constant and variable. Abu-Nada et al. [82,83,84,85] introduced a cycle model with variable SH of WF with the nonlinear function (NLF) of temperature in the performance studies of internal combustion engine cycle. Using the variable SH model introduced in references [82,83,84,85], references [86,87,88,89] studied the performances of Otto [86], Diesel [87], Atkinson [88] and Dual [89] cycles and analyzed the impacts of loss items on cycle PC.

In the third class, considering the variable SHR of WF with the LF of temperature, Ebrahimi established endoreversible [90] and irreversible [91] Dual cycle, endoreversible Atkinson cycle [92], endoreversible Diesel cycle [93] and irreversible Otto cycle [94] models, and analyzed the impacts of variable SHR and loss items on cycle PC. Moreover, considering the variable SHR of WF with NLF of temperature, references [95,96,97,98] proposed the endoreversible Diesel cycle [95], irreversible Atkinson cycle [96], endoreversible [97] and irreversible [98] Dual cycle models, and studied the effects of variable SHR and loss items on cycle PC.

Establishing the universal model and obtaining the universal laws and results are the aims of FTT pursuit, and they are the same for a performance study of RHEC cycle. For the generalized irreversible RHEC model established in references [73,81], there is no work in the open literature which has studied the effect of variable SHR of WF with NLF of temperature on RHEC PC. This paper will combine the generalized irreversible RHEC model established in [73,81] and the variable SHR model of WF with NLF of temperature in [95,96,97,98], analyze the outcome and compare the effects of various SHR models on cycle PC. The maximum power outputs (MPOs) and the corresponding optimal compression ratios, as well as the maximum efficiencies and the corresponding optimal compression ratios for various special cycles with three SHR models will be also compared. 

## 2. Cycle Model

An AS RHEC model is shown in Figure 1 which contains two adiabatic branches; two endothermic processes with SH of $C_{in1}$ and $C_{in2}$; and two exothermic processes with SH of $C_{out1}$ and $C_{out2}$, respectively.

When the above four SH are different values, this universal cycle model will be simplified to all kinds of special cycle models. The two irreversible adiabatic processes are shown as 1→2 and 4→5; the two heating processes are shown as 2→3 and; and the two cooling processes are shown as 5→6 and 6→1.

Assuming the SHR of WF varied with temperature with NLF, the SHR γ can be written as
(1)$$\gamma =c+bT+aT^{2}$$
where a, b and c are constants and T is WF temperature. 

It can be supposed that the four thermal capacities of the cycle are
(2)$$\begin{array}{cc} C_{in1}=R\frac{m_{in1}(aT^{2}+bT)+n_{in1}}{aT^{2}+bT+c-1} & C_{in2}=R\frac{m_{in2}(aT^{2}+bT)+n_{in2}}{aT^{2}+bT+c-1}\\ C_{out1}=R\frac{m_{out1}(aT^{2}+bT)+n_{out1}}{aT^{2}+bT+c-1} & C_{out2}=R\frac{m_{out2}(aT^{2}+bT)+n_{out2}}{aT^{2}+bT+c-1} \end{array}$$
where $m_{in1}$, $m_{in2}$, $m_{out1}$, $m_{out2}$, $n_{in1}$, $n_{in2}$, $n_{out1}$ and $n_{out2}$ are constants. When $m_{in1}$ is 1, $n_{in1}$ is c, and when $m_{in1}$ is 0, $n_{in1}$ is 1, so do $m_{in2}$ and $n_{in2}$, $m_{out1}$ and $n_{out1}$ and $m_{out2}$ and $n_{out2}$.

When the constants have different values, the four thermal capacities can change into the SH with constant pressure and constant volume
(3)$$C_{P}=\frac{R(aT^{2}+bT+c)}{aT^{2}+bT+c-1},C_{V}=\frac{R}{aT^{2}+bT+c-1}$$

It can be supposed that two adiabatic processes are instantaneous, and the temperature of WF changes at a constant speed. $k_{1}$, $k_{2}$, $k_{3}$ and $k_{4}$ are constants, then the time spent on each cycle is
(4)$$\begin{array}{c} \tau =t_{out2}+t_{out1}+t_{in2}+t_{in1}\\ =k_{4}(T_{6}-T_{1})+k_{3}(T_{5}-T_{6})+k_{2}(T_{4}-T_{3})+k_{1}(T_{3}-T_{2}) \end{array}$$

The heat addition in the processes 2→3 and 3→4 can be written as
(5)$$\begin{array}{c} Q_{in}=M\left(\int_{T_{2}}^{T_{3}}C_{in1}dT+\int_{T_{3}}^{T_{4}}C_{in2}dT\right)\\ =MR\left\{\begin{array}{l} m_{in1}(T_{3}-T_{2})+m_{in2}(T_{4}-T_{3})+[2D^{-1}(m_{in1}-cm_{in1}+n_{in1})]\{\arctan[D^{-1}(b+2aT_{3})]-\arctan[D^{-1}\\ \left(b+2aT_{2})]\}-\{\arctan[D^{-1}(b+2aT_{3})]-\arctan[D^{-1}(b+2aT_{4})]\}\times [2D^{-1}(m_{in2}-cm_{in2}+n_{in2})\right] \end{array}\right\} \end{array}$$

The heat rejection in the processes 5→6 and 6→1 can be written as
(6)$$\begin{array}{c} Q_{out}=M\left(\int_{T_{1}}^{T_{6}}C_{out1}dT+\int_{T_{6}}^{T_{5}}C_{out2}dT\right)\\ =MR\left\{\begin{array}{l} m_{out1}(T_{6}-T_{1})+m_{out2}(T_{5}-T_{6})+[2D^{-1}(m_{out1}-cm_{out1}+n_{out1})]\{\arctan[D^{-1}(b+2aT_{6})]-\arctan[D^{-1}\\ \left(b+2aT_{1})]\}-\{\arctan[D^{-1}(b+2aT_{6})]-\arctan[D^{-1}(b+2aT_{5})]\}\times [(2D^{-1}(m_{out2}-cm_{out2}+n_{out2}))\right] \end{array}\right\} \end{array}$$
where $D=\sqrt{4ac-4a-b^{2}}$, R is gas constant and M is mole number of WF.

The following parameters are defined as
(7)$$r=V_{1}/V_{2},\rho =V_{4}/V_{3},r_{p}=T_{3}/T_{2},r_{c}=T_{6}/T_{1}$$

For the two irreversible adiabatic processes, the IIL are defined as the expansion and compression efficiencies [86,87,88,89]
(8)$$\eta_{C}=\left(T_{1}-T_{2S}\right)/\left(T_{1}-T_{2}\right)$$
(9)$$\eta_{E}=\left(T_{5}-T_{4}\right)/\left(T_{5S}-T_{4}\right)$$

According to references [75,76,77,78,79,80,81], the expression for reversible adiabatic process when SHR is varied is
(10)$$V^{\gamma -1}T={\left(V+dV\right)}^{\gamma -1}(T+dT)$$

From Equation (10), one has
(11)$$\frac{1}{c-1}\left\{\frac{1}{2}\ln\left(\frac{aT_{j}^{2}+bT_{j}+c-1}{aT_{i}^{2}+bT_{i}+c-1}\right)+\frac{b}{D}\left\{\begin{array}{l} \arctan[(2aT_{j}+b)/D]\\ -\arctan[(2aT_{i}+b)/D] \end{array}\right\}-\ln\left(\frac{T_{j}}{T_{i}}\right)\right\}=\ln\left(\frac{V_{j}}{V_{i}}\right)$$

For the endoreversible adiabatic process 1→2S, one has
(12)$$\frac{1}{c-1}\left\{\frac{1}{2}\ln\left(\frac{aT_{2S}^{2}+bT_{2S}+c-1}{aT_{1}^{2}+bT_{1}+c-1}\right)+\frac{b}{D}\left\{\begin{array}{l} \arctan[(2aT_{2S}+b)/D]\\ -\arctan[(2aT_{1}+b)/D] \end{array}\right\}-\ln\left(\frac{T_{2S}}{T_{1}}\right)\right\}=\ln\left(\frac{1}{r}\right)$$

The special cycles $m_{in1}$, $m_{in2}$, $m_{out1}$, $m_{out2}$, $n_{in1}$, $n_{in2}$, $n_{out1}$ and $n_{out2}$ are fixed, and Equation (12) becomes the expression of the adiabatic process for the various special cycles.

After a cycle, the entropy change of the WF is zero, so one has
(13)$$\left\{\begin{array}{l} m_{in1}\ln(T_{3}/T_{2S})+m_{in2}\ln(T_{4}/T_{3})+m_{out1}\ln(T_{1}/T_{6})+m_{out2}\ln(T_{6}/T_{5S})+\{[m_{in1}(c-1)\\ -n_{in1}]{\left[2(c-1)\right]}^{-1}\}\{D^{-1}2b\{-\arctan[D^{-1}(b+2aT_{2S})+\arctan[D^{-1}(b+2aT_{3})]]\}-2\\ \times \ln(T_{3}/T_{2S})+\ln[(a{T_{3}}^{2}+bT_{3}+c-1)/(a{T_{2S}}^{2}+bT_{2S}+c-1)]\}+\{[m_{in2}(c-1)\\ -n_{in2}]{\left[2(1-c)\right]}^{-1}\}\{D^{-1}2b\{-\arctan[D^{-1}(b+2aT_{3})]+\arctan[D^{-1}(b+2aT_{4})]\}-2\\ \times \ln(T_{4}/T_{3})+\ln[(a{T_{4}}^{2}+bT_{4}+c-1)/(a{T_{3}}^{2}+bT_{3}+c-1)]\}-\{[m_{out1}(c-1)\\ -n_{out1}]{\left[2(1-c)\right]}^{-1}\}\{D^{-1}2b\{-\arctan[D^{-1}(b+2aT_{6})]+\arctan[D^{-1}(b+2aT_{1})]\}-2\\ \times \ln(T_{1}/T_{6})+\ln[(aT_{1}^{2}+bT_{1}+c-1)/(a{T_{6}}^{2}+bT_{6}+c-1)]\}-\{[m_{out2}(c-1)\\ -n_{out2}]{\left[2(1-c)\right]}^{-1}\}\{D^{-1}2b\{-\arctan[D^{-1}(b+2aT_{5S})]+\arctan[D^{-1}(b+2aT_{6})]\}-2\\ \times \ln(T_{6}/T_{5S})+\ln[(a{T_{6}}^{2}+bT_{6}+c-1)/(a{T_{5S}}^{2}+bT_{5S}+c-1)]\} \end{array}\right\}=0$$

For a practical cycle, there exists HTL and FL. According reference [67], the heat addition rate to the WF by combustion is:(14)$$Q_{in}=A^{\prime }-B^{\prime }[0.5(T_{4}+T_{2})-T_{0}]=A-B(T_{4}+T_{2})$$
where $A^{\prime }$ is the heat released by fuel, $B^{\prime }$ is heat leakage coefficient, $T_{0}$ is the ambient temperature and $A=A^{\prime }+B^{\prime }T_{0}$ and $B=B^{\prime }/2$ are two constants.

According to reference [70], the lost power due to FL is
(15)$$P_{\mu }=\mu {\left(dX/dt\right)}^{2}=\mu {\bar{v}}^{2}$$

The mean velocity of piston motion is
(16)$$\bar{v}=x(r-1)/\Delta t_{12}$$
where μ is the friction coefficient in exhaust stroke, $\bar{v}$ is the mean velocity of piston, X is the piston displacement, x is the piston position at upper dead point and $\Delta t_{12}$ is the time of the power stroke.

## 3. Power Output and Thermal Efficiency

The net power output is
(17)$$\begin{array}{c} P=(Q_{in}-Q_{out})/\tau -P_{\mu }\\ =\frac{MR\left\{\begin{array}{l} m_{in1}(T_{3}-T_{2})+m_{in2}(T_{4}-T_{3})-m_{out1}(T_{6}-T_{1})-m_{out2}(T_{5}-T_{6})+[2D^{-1}(m_{in1}-cm_{in1}\\ +n_{in1})]\{\arctan[D^{-1}(b+2aT_{3})]-\arctan[D^{-1}(b-+2aT_{2})]\}+[2D^{-1}(m_{in2}-cm_{in2}\\ +n_{in2})]\{\arctan[D^{-1}(b+2aT_{4})]-\arctan[D^{-1}(b+2aT_{3})]\}-[2(m_{out1}-cm_{out1}+\\ n_{out1})/D]\{\arctan[D^{-1}(b+2aT_{6})]-\arctan[D^{-1}(b+2aT_{1})]\}+[2D^{-1}(m_{out2}-cm_{out2}\\ +n_{out2})]\{\arctan[D^{-1}(b+2aT_{6})]-\arctan[D^{-1}(b+2aT_{5})]\} \end{array}\right\}}{K_{4}(T_{6}-T_{1})+K_{3}(T_{5}-T_{6})+K_{2}(T_{4}-T_{3})+K_{1}(T_{3}-T_{2})}-b_{f}{\left(r-1\right)}^{2} \end{array}$$
where $b_{f}$ is defined as $b_{f}=\mu {x_{2}}^{2}/{\left(\Delta t_{12}\right)}^{2}$.

The thermal efficiency is
(18)$$\begin{array}{c} \eta =P\tau /Q_{in}\\ =\frac{\begin{array}{l} MR\left\{\begin{array}{l} m_{in1}(T_{3}-T_{2})+m_{in2}(T_{4}-T_{3})-m_{out1}(T_{6}-T_{1})-m_{out2}(T_{5}-T_{6})-[2D^{-1}(m_{in1}-cm_{in1}+n_{in1})]\\ \left\{\arctan[D^{-1}(b+2aT_{2})]-\arctan[D^{-1}(b+2aT_{3})]\}-[2D^{-1}(m_{in2}-cm_{in2}+n_{in2})\right]\\ \left\{\arctan[D^{-1}(b+2aT_{3})]-\arctan[D^{-1}(b+2aT_{4})]\}+[2D^{-1}(m_{out1}-cm_{out1}+n_{out1})\right]\\ \left\{\arctan[D^{-1}(b+2aT_{1})]-\arctan[D^{-1}(b+2aT_{6})]\}+[2D^{-1}(m_{out2}-cm_{out2}+n_{out2})\right]\\ \left\{\arctan[D^{-1}(b+2aT_{6})]-\arctan[D^{-1}(b+2aT_{5})]\right\} \end{array}\right\}\\ -b_{f}{\left(r-1\right)}^{2}[K_{4}(T_{6}-T_{1})+K_{3}(T_{5}-T_{6})+K_{2}(T_{4}-T_{3})+K_{1}(T_{3}-T_{2})] \end{array}}{MR\left\{\begin{array}{l} m_{in1}(T_{3}-T_{2})+m_{in2}(T_{4}-T_{3})-[2D^{-1}(m_{in1}-cm_{in1}+n_{in1})]\{\arctan[D^{-1}(b+2aT_{2})]-\\ \arctan[D^{-1}(b+2aT_{3})]\}-[2D^{-1}(m_{in2}-cm_{in2}+n_{in2})]\{\arctan[D^{-1}(b+2aT_{3})]-\\ \arctan[D^{-1}(b+2aT_{4})]\} \end{array}\right\}} \end{array}$$

In order to make the cycle run normally, State 3 must be between States 2 and 4. When States 2 and 3 coincide, it gives $r_{p}={\left(r_{p}\right)}_{\min}=1$, and when States 3 and 4 coincide, it gives $r_{p}={\left(r_{p}\right)}_{\max}$ and $T_{3}={\left(r_{p}\right)}_{\max}T_{2}=T_{4}$. $T_{2}$ can be gotten by Equations (12) and (8). Substituting $T_{3}={\left(r_{p}\right)}_{\max}T_{2}=T_{4}$ into Equations (5) and (14) gives ${\left(r_{p}\right)}_{\max}$. So the range of $r_{p}$ is
(19)$$1={\left(r_{p}\right)}_{\min}\leq r_{p}\leq {\left(r_{p}\right)}_{\max}$$

State 6 must be between States 1 and 5. When States 1 and 6 coincide, it gives $r_{c}={\left(r_{c}\right)}_{\min}=1$, and when States 3 and 4 coincide, it gives $r_{c}={\left(r_{c}\right)}_{\max}$ and $T_{6}={\left(r_{c}\right)}_{\max}T_{1}=T_{5}$. Substituting $T_{2}$ into Equation (7) gives $T_{3}$, substituting $T_{3}$ into Equations (5) and (14) gives $T_{4}$ and substituting $T_{6}={\left(r_{c}\right)}_{\max}T_{1}=T_{5}$ and the temperatures above into Equation (13) gives ${\left(r_{p}\right)}_{\max}$. So the range of $r_{c}$ is
(20)$$1={\left(r_{c}\right)}_{\min}\leq r_{c}\leq {\left(r_{c}\right)}_{\max}$$

## 4. Discussions

Equations (17) and (18) are the PAE characteristics of the universal cycle which include all kinds of RHEC with different loss items. 

(1) When $m_{in1}=m_{in2}=m_{out1}=m_{out2}=0$ and $n_{in1}=n_{in2}=n_{out1}=n_{out2}=1$, the expressions can be simplified into the PAE for an AS Otto cycle.

(2) When $m_{in1}=m_{in2}=1$, $n_{in1}=n_{in2}=c$, $m_{out1}=m_{out2}=0$ and $n_{out1}=n_{out2}=1$, the expressions can be simplified into the PAE for an AS Diesel cycle.

(3) When $m_{in1}=m_{in2}=0$, $n_{in1}=n_{in2}=1$, $m_{out1}=m_{out2}=1$ and $n_{out1}=n_{out2}=c$, the expressions can be simplified into the PAE for an AS Atkinson cycle.

(4) When $m_{in1}=m_{in2}=m_{out1}=m_{out2}=1$ and $n_{in1}=n_{in2}=n_{out1}=n_{out2}=c$, the expressions can be simplified into the PAE for an AS Brayton cycle.

(5) When $m_{in1}=0$, $m_{in2}=m_{out1}=m_{out2}=n_{in1}=1$ and $n_{in2}=n_{out1}=n_{out2}=c$, the expressions can be simplified into the PAE for an AS Dual cycle.

(6) When $m_{in1}=m_{in2}=m_{out2}=0$, $m_{out1}=n_{in1}=n_{in2}=n_{out2}=1$ and $n_{out1}=c$, the expressions can be simplified into the PAE for an AS Miller cycle.

(7) When a≠0, the expressions can be simplified into the PAE for the universal cycle with variable SHR of WF with the NLF of temperature; when a=0 and b≠0, the expressions can be turned into those with variable SHR with the LF of temperature; and when a=b=0, the expressions can be simplified into those for the constant SHR.

(8) When $\eta_{c}\neq 1$ and $\eta_{e}\neq 1$, the expressions can be simplified into the PAE for the universal cycle with IIL, and when $\eta_{c}=\eta_{e}=1$, the expressions can be simplified into that for the cycle without IIL.

(9) When $b_{f}\neq 0$, the expressions can be simplified into the PAE for the cycle with FL, and when $b_{f}=0$, the expressions can be simplified into those for the cycle without FFL.

(10) When B≠0, the expressions can be simplified into the PAE for the cycle with HTL, and when B=0, the expressions can be simplified into those for the cycle without HTL.

## 5. Numerical Examples

According to references [81,95,96,97,98], the following constants are used in the computations: A=60000 J/mol, B=25 J/(mol⋅K), $T_{1}=300\;K$, $b=-9.7617\times 10^{-5}\;K^{-1}$, $a=1.6928\times 10^{-8}\;K^{-1}$, c=1.4235, M=0.0157 mol, R=8.314, $\gamma_{p}=1.2$, $\gamma_{c}=1.2$, $b_{f}=32.5\;W$, $K_{3}=K_{4}=18.67\times 10^{-6}\;s\cdot K^{-1}$ and $K_{1}=K_{2}=8.128\times 10^{-6}\;s\cdot K^{-1}$.

Figure 2 and Figure 3 illustrate the relations of power output versus CR and efficiency versus CR for various special cycles with the constant SHR and variable SHR with LF and NLF of temperature. Figure 2 shows that, compared with the constant SHR, the ranges of the CRs of various special cycles with the variable SHR with the LF of temperature increase from 13.5 to about 16, the power output decreases (0.5% decrease for Miller cycle; 3% decrease for Otto and Atkinson cycles; 8% decrease for Diesel, Brayton and Dual cycles). Compared with the variable SHR with the LF of temperature, the ranges of CR of various special cycles with the variable SHR with NLF of temperature decrease from 16 to about 15; the changes of power output are unobvious (0.5% increase for Diesel cycle; 0.1% increase for Dual cycle; about 0.5–2% decrease for Otto, Atkinson, Brayton and Miller cycles). The order of the maximum power outputs (MPO) of various special cycles is $P_{br}>P_{at}>P_{mi}>P_{du}>P_{di}>P_{ot}$ with every one of the three SHR models. 

For the Otto cycle, under three SHR models, the orders of the MPOs and the corresponding optimal CRs are ${\left(P_{ot}\right)}_{C}>{\left(P_{ot}\right)}_{L}>{\left(P_{ot}\right)}_{N}$ and ${\left(r_{ot}\right)}_{L}>{\left(r_{ot}\right)}_{N}>{\left(r_{ot}\right)}_{C}$. For the Diesel cycle, under three SHR models, the orders of the MPOs and the corresponding optimal CRs are ${\left(P_{di}\right)}_{C}>{\left(P_{di}\right)}_{N}>{\left(P_{di}\right)}_{L}$ and ${\left(r_{di}\right)}_{C}>{\left(r_{di}\right)}_{L}>{\left(r_{di}\right)}_{N}$. For Atkinson cycle, under three SHR models, the orders of the MPOs and the corresponding optimal CRs are ${\left(P_{at}\right)}_{C}>{\left(P_{at}\right)}_{L}>{\left(P_{at}\right)}_{N}$ and ${\left(r_{at}\right)}_{L}>{\left(r_{at}\right)}_{N}>{\left(r_{at}\right)}_{C}$. For the Brayton cycle, under three SHR models, the orders of the MPOs and the corresponding optimal CRs are ${\left(P_{br}\right)}_{C}>{\left(P_{br}\right)}_{L}>{\left(P_{br}\right)}_{N}$ and ${\left(r_{br}\right)}_{L}>{\left(r_{br}\right)}_{C}>{\left(r_{br}\right)}_{N}$. For the Dual cycle, under three SHR models, the orders of the MPOs and the corresponding optimal CRs are ${\left(P_{du}\right)}_{C}>{\left(p_{du}\right)}_{L}>{\left(P_{du}\right)}_{N}$ and ${\left(r_{du}\right)}_{C}>{\left(r_{du}\right)}_{L}>{\left(r_{du}\right)}_{N}$. For the Miller cycle, under three SHR models, the orders of the MPOs and the corresponding optimal CRs are ${\left(P_{mi}\right)}_{C}>{\left(P_{mi}\right)}_{L}>{\left(P_{mi}\right)}_{N}$ and ${\left(r_{mi}\right)}_{L}>{\left(r_{mi}\right)}_{N}>{\left(r_{mi}\right)}_{C}$. Under constant SHR model, the order of the corresponding optimal CRs at the MPO points of various special cycles is $r_{di}>r_{du}>r_{br}>r_{mi}>r_{ot}>r_{at}$. Under variable SHR with the LF of the temperature model, the order of the corresponding optimal CRs at the MPO points of various special cycles is $r_{di}>r_{br}>r_{mi}>r_{du}>r_{ot}>r_{at}$. Under variable SHR with NLF of temperature model, the order of the corresponding optimal CRs at the MPO points of various special cycles is $r_{di}>r_{du}>r_{mi}>r_{br}>r_{ot}>r_{at}$.

Figure 3 shows that, compared with the constant SHR, the efficiencies of various special cycles with the variable SHR with the LF of temperature decrease by about 12%. Compared with the variable SHR with the LF of temperature, the efficiencies of various special cycles with variable SHR with NLF of temperature increase by about 0.7–1.7%. The order of the maximum efficiency of various special cycles is $\eta_{at}>\eta_{mi}>\eta_{br}>\eta_{ot}>\eta_{du}>\eta_{di}$ with every one of the three SHR models. 

For the Otto cycle, under three SHR models, the orders of the maximum efficiencies and corresponding optimal CRs are ${\left(\eta_{ot}\right)}_{C}>{\left(\eta_{ot}\right)}_{N}>{\left(\eta_{ot}\right)}_{L}$ and ${\left(r_{ot}\right)}_{C}>{\left(r_{ot}\right)}_{L}>{\left(r_{ot}\right)}_{N}$. For the Diesel cycle, under three SHR models, the orders of the maximum efficiencies and corresponding optimal CRs are ${\left(\eta_{di}\right)}_{C}>{\left(\eta_{di}\right)}_{N}>{\left(\eta_{di}\right)}_{L}$ and ${\left(r_{di}\right)}_{C}>{\left(r_{di}\right)}_{N}>{\left(r_{di}\right)}_{L}$. For the Atkinson cycle, under three SHR models, the orders of the maximum efficiencies and corresponding optimal CRs are ${\left(\eta_{at}\right)}_{C}>{\left(\eta_{at}\right)}_{N}>{\left(\eta_{at}\right)}_{L}$ and ${\left(r_{at}\right)}_{L}>{\left(r_{at}\right)}_{N}>{\left(r_{at}\right)}_{C}$. For the Brayton cycle, under three SHR models, the orders of the maximum efficiencies and corresponding optimal CRs are ${\left(\eta_{br}\right)}_{C}>{\left(\eta_{br}\right)}_{N}>{\left(\eta_{br}\right)}_{L}$ and ${\left(r_{br}\right)}_{C}>{\left(r_{br}\right)}_{N}>{\left(r_{br}\right)}_{L}$. For the Dual cycle, under three SHR models, the orders of the maximum efficiencies and corresponding optimal CRs are ${\left(\eta_{du}\right)}_{C}>{\left(\eta_{du}\right)}_{N}>{\left(\eta_{du}\right)}_{L}$ and ${\left(r_{du}\right)}_{C}>{\left(r_{du}\right)}_{N}>{\left(r_{du}\right)}_{L}$. For the Miller cycle, under three SHR models, the orders of the maximum efficiencies and corresponding optimal CRs are ${\left(\eta_{mi}\right)}_{C}>{\left(\eta_{mi}\right)}_{N}>{\left(\eta_{mi}\right)}_{L}$ and ${\left(r_{mi}\right)}_{L}>{\left(r_{mi}\right)}_{N}>{\left(r_{mi}\right)}_{C}$. 

Under the constant SHR model, the order of the corresponding optimal CRs at the maximum efficiency points of various special cycles is $r_{di}>r_{du}>r_{br}>r_{ot}>r_{mi}>r_{at}$. Under variable SHR with the LF of temperature model, the order of the corresponding optimal CRs at the maximum efficiency points of various special cycles is $r_{di}>r_{du}>r_{br}>r_{mi}>r_{ot}>r_{at}$. Under variable SHR with NLF of temperature model, the order of the corresponding optimal CRs at the maximum efficiency points of various special cycles is $r_{di}>r_{du}>r_{br}>r_{ot}>r_{mi}>r_{at}$.

In general, the optimal CR at MPO point is not the same as the optimal CR at maximum efficiency point, for all discussed cycles with three SHR models. The reasonable design range for all of the discussed cycles with three SHR models should be between the optimal CR at MPO point and the optimal CR at maximum efficiency point from the point of view of compromised optimization of the PAE. 

From what was mentioned above, one can see that there are influences of the variable SHR model on the performance of every special cycle; and the performances of Miller, Brayton and Atkinson cycles are more excellent than those of Otto, Diesel and Dual cycles with every one of the three SHR models. 

## 6. Conclusions

The AS RHEC model considering HTL, FL and IIL is established in this paper. The cycle performances with various SHR are analyzed. The performance parameters including the PAE are derived. The performances of all kinds of special cycles are discussed and the MPO and the maximum efficiency of each special cycle and the corresponding optimal CRs are compared. The results show that the orders of the MPO and the maximum efficiency remain the same with every one of the three SHR models, but the PAE changes, which suggests that the various SHRs have influences on cycle performance. When the model of variable SHR is more complicated, the distance between the cycle model and the practice one is closer. The reasonable design range for various cycles should be between the optimal CR at MPO point and the optimal CR at maximum efficiency point for the compromise optimization of the PAE. 

## Figures and Tables

**Figure 1 entropy-22-00397-f001:**
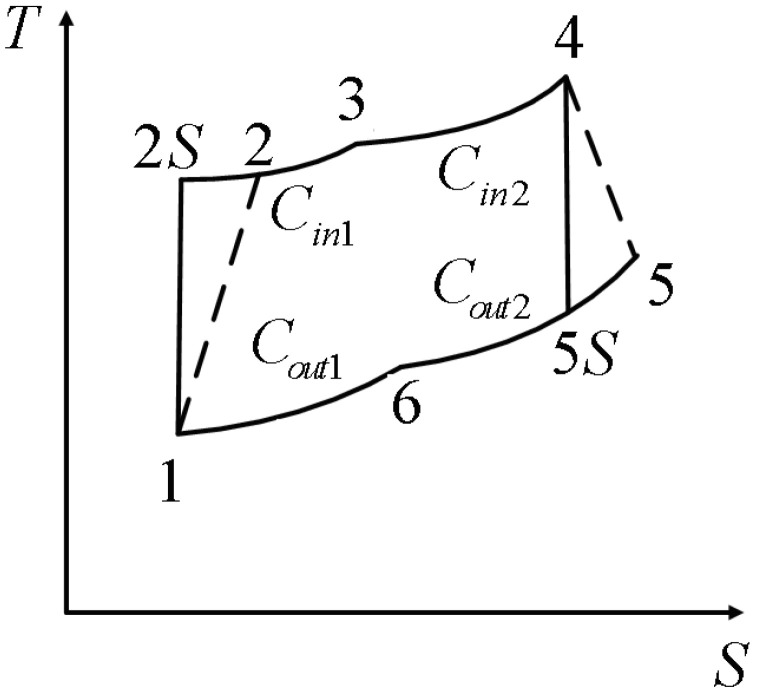
The T−s diagram for an irreversible reciprocating heat-engine cycle (RHEC) model.

**Figure 2 entropy-22-00397-f002:**
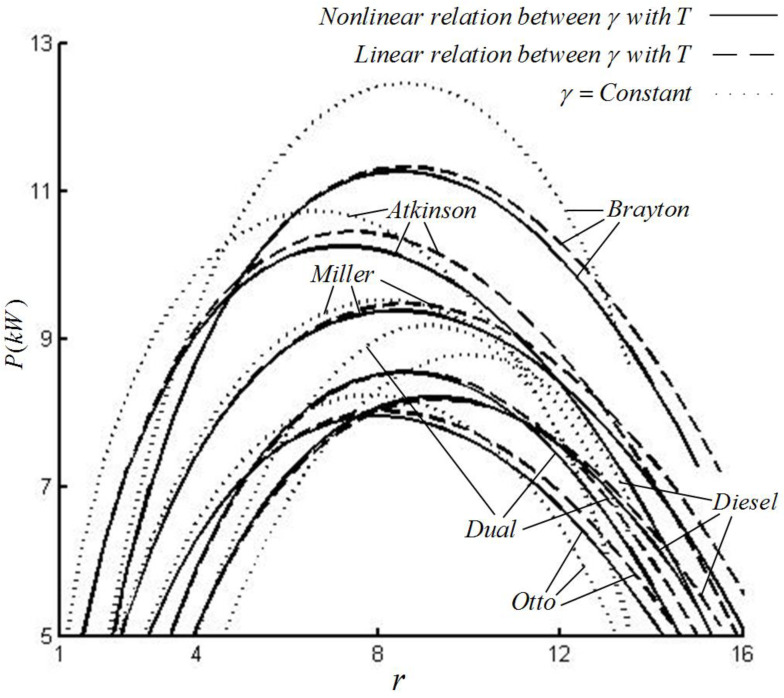
The power output versus CR for various special cycles.

**Figure 3 entropy-22-00397-f003:**
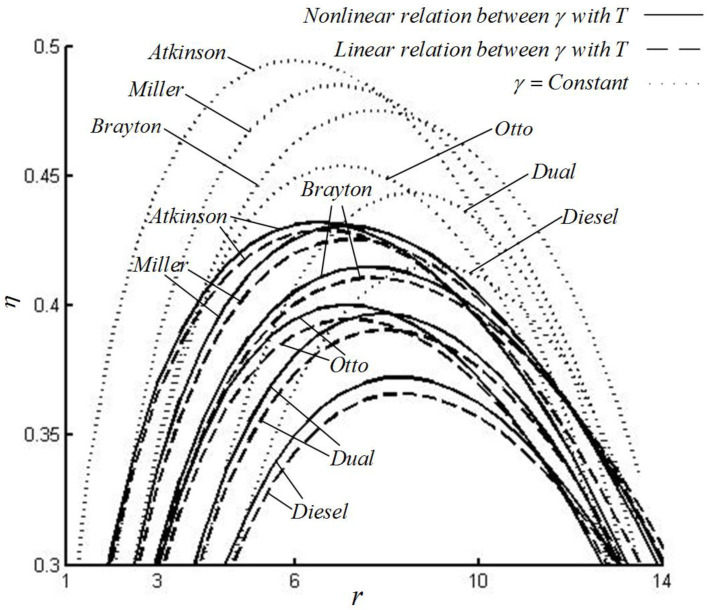
The efficiency versus compression ratio (CR) for various special cycles.

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
