# Peer review of "Performance of Universal Reciprocating Heat-Engine Cycle with Variable Specific Heats Ratio of Working Fluid"

_entropy, 2020, doi:10.3390/e22040397_

Round 1

Reviewer 1 Report

The article "Performance of Universal Reciprocating Heat-Engine Cycle with Variable Specific Heats Ratio of Working Fluid" by Chen et al. presents the study on the performance of the generalized reciprocating heat-engine cycle using the finite-time thermodynamics. The authors provided a theoretical derivation of the power and efficiency of a universal engine cycle and performed a numerical simulation to compare them of various individual cycles.

Over the past several decades, there has been an intense theoretical effort to understand the optimization of the power and efficiency of the engine cycle, and I find the results presented in the manuscript are interesting and look to be valid to me. Therefore, I would like to see this published in Entropy. However, I would like to ask what the significant contribution of this paper is. The answer is needed in the introduction of the manuscript.

Minor comment:
1) In line 21, What do you mean by "it"?
2) In line 150-152, the definitions of the variables don't match with the previous ones.

Author Response

In General

All of the comments were accepted, and the manuscript was modified. All revisions related to the reviewers’ comments were highlighted by red words in the revised manuscript.

The following sentence was added in the “Acknowledgements”:

“The authors wish to thank the reviewers for their careful, unbiased and constructive suggestions, which led to this revised manuscript.”

Reply to the comment of Reviewer 1

Comment 1: I would like to ask what the significant contribution of this paper is. The answer is needed in the introduction of the manuscript.

Accepted: the following sentences have been added in the introduction of the manuscript.

“In the early studies, for the reciprocating heat-engine cycle (RHEC), the specific heats (SH) of WF was usually assumed to be constant. For practical cycle, the property and composition of the WF will change with the happening of combustion reaction. So the SH of WF will also change with the happening of combustion reaction and this change will has great influence on cycle performance. The variation of SH of WF would inevitably cause the variation of SHR.”

“Establishing the universal model and obtaining the universal laws and results are the aim of FTT pursuing, and that is the same for the performance study of RHEC cycle. For the generalized irreversible RHEC model established in Refs. [73, 81], there is no work in the open literature which has studied the effect of variable SHR of WF with NLF of temperature on RHEC PC.”

Comment 2: In line 21, What do you mean by "it"?

Accepted and Explanation: in line 21, “it” means specific heat ratio (SHR) and “it” has been substituted by “the SHR” in revised version

Comment 3: In line 150-152, the definitions of the variables don't match with the previous ones.

Explanation: in lines 150-152, the symbols of the variables may be not the same as those of in the previous papers, but the definitions and physical meanings of the variations are the same as the previous references [81, 95-98].

Reviewer 2 Report

This paper deals with performance of universal reciprocating heat-engine cycle with variable specific heats ratio of working fluid. The introduction presents a good summary of previous work on the subject. After this introduction the authors describe cycle model. After this, the authors pass to the results. There is only two figures for the results and also only twenty lines for the comments on the figures.

In my opinion this paper cannot be accepted in the present form for publication in Entropy.

the paper should be improved before be accepted for publication.

Author Response

In General

All of the comments were accepted, and the manuscript was modified. All revisions related to the reviewers’ comments were highlighted by red words in the revised manuscript.

The following sentence was added in the “Acknowledgements”:

“The authors wish to thank the reviewers for their careful, unbiased and constructive suggestions, which led to this revised manuscript.

Reply to the comments of Reviewer 2:

Comment 1: There is only two figures for the results and also only twenty lines for the comments on the figures. The paper should be improved before be accepted for publication.

Accepted: The paper was improved:

  1. The subscripts have been added in Nomenclature.
  2. The following sentences were added in the Abstract:

“The results include the PAE performance characteristics of various special cycles (including Miller, Dual, Atkinson, Brayton, Diesel, and Otto cycles) when the SHR of WF is constant and variable (including the SHR varied with linear function (LF) and nonlinear function (NLF) of temperature). The maximum power outputs and the corresponding optimal compression ratios, as well as the maximum efficiency and the corresponding optimal compression ratios for various special cycles with three SHR models are compared.”

  1. The discussions about the Fig. 2 were improved, and the following sentences were added in the section of 5. Numerical Examples:

For Otto cycle, under three SHR models, the orders of the MPOs and the corresponding optimal CRs are  and . For Diesel cycle, under three SHR models, the orders of the MPOs and the corresponding optimal CRs are  and . For Atkinson cycle, under three SHR models, the orders of the MPOs and the corresponding optimal CRs are  and . For Brayton cycle, under three SHR models, the orders of the MPOs and the corresponding optimal CRs are  and . For Dual cycle, under three SHR models, the orders of the MPOs and the corresponding optimal CRs are  and . For Miller cycle, under three SHR models, the orders of the MPOs and the corresponding optimal CRs are  and .

Under constant SHR model, the order of the corresponding optimal CRs at the MPO points of various special cycles is . Under variable SHR with LF of temperature model, the order of the corresponding optimal CRs at the MPO points of various special cycles is . Under variable SHR with NLF of temperature model, the order of the corresponding optimal CRs at the MPO points of various special cycles is .

  1. The discussions about the Fig. 3 were improved, and the following sentences were added in the section of 5. Numerical Examples:

 “For Otto cycle, under three SHR models, the orders of the maximum efficiencies and corresponding optimal CRs are  and. For Diesel cycle, under three SHR models, the orders of the maximum efficiencies and corresponding optimal CRs are  and . For Atkinson cycle, under three SHR models, the orders of the maximum efficiencies and corresponding optimal CRs are  and . For Brayton cycle, under three SHR models, the orders of the maximum efficiencies and corresponding optimal CRs are  and . For Dual cycle, under three SHR models, the orders of the maximum efficiencies and corresponding optimal CRs are  and. For Miller cycle, under three SHR models, the orders of the maximum efficiencies and corresponding optimal CRs are  and .

Under constant SHR model, the order of the corresponding optimal CRs at the maximum efficiency points of various special cycles is . Under variable SHR with LF of temperature model, the order of the corresponding optimal CRs at the maximum efficiency points of various special cycles is . Under variable SHR with NLF of temperature model, the order of the corresponding optimal CRs at the maximum efficiency points of various special cycles is .

  1. The discussions about the Figs. 2 and 3 were improved, and the following sentences were added in the section of 5. Numerical Examples:

In general, the optimal CR at MPO point is not the same as the optimal CR at maximum efficiency point, for all discussed cycles with three SHR models. The reasonable design range for all of the discussed cycles with three SHR models should be between the optimal CR at MPO point and the optimal CR at maximum efficiency point from the point of view of compromise optimization of the PAE.

  1. The following sentence was added in the section of 6. Conclusion:

The reasonable design range for various cycles should be between the optimal CR at MPO point and the optimal CR at maximum efficiency point for the compromise optimization of the PAE.

Reviewer 3 Report

The performance of a reciprocating heat-engine cycle model is determined, obtaining plots (i.e., Figure 2 and 3) of the power output and efficiency in terms of the compression ratio, assuming different dependences of the specific heat ratio on temperature. The article is clear and interesting, so it deserves to be published.

My only concern is that the English should be improved. Here in the following a few typos are listed.

Page 1, line 13: “found”, instead of “founded”.

Page 1, line 14: “consists of”, instead of “is consisted of”.

Page 1, the last sentence of the Abstract must to be rephrased.

Page 2, line 2: processes, a series of achievements were made, including….

Page 5, line 116, 119 and 120:  “coincides”, instead of “are coincided”

Page 8, last line: the more complicated the model of variable SHR, the closer the distance between model and actual cycle.

Author Response

In General

All of the comments were accepted, and the manuscript was modified. All revisions related to the reviewers’ comments were highlighted by red words in the revised manuscript.

The following sentence was added in the “Acknowledgements”:

“The authors wish to thank the reviewers for their careful, unbiased and constructive suggestions, which led to this revised manuscript.”

Reply to the comments of Reviewer 3:

Comment 1: Page 1, line 13: “found”, instead of “founded”.

Explanation: in line 13, the sentence “an air standard reciprocating heat-engine cycle model is founded by using finite time thermodynamics” should be the passive voice, so the verb “found” should use the past participle “founded”

Comment 2: Page 1, line 14: “consists of”, instead of “is consisted of”.

Accepted: in line 14, “is consisted of” has been substituted by “consists of”

Comment 3: Page 1, the last sentence of the Abstract must to be rephrased.

Accepted: the last sentence of Abstract has been rephrased as

“The results include the PAE performance characteristics of various special cycles (including Miller, Dual, Atkinson, Brayton, Diesel, and Otto cycles) when the SHR of WF is constant and variable (including the SHR varied with linear function (LF) and nonlinear function (NLF) of temperature).”

Comment 4: Page 2, line 2: processes, a series of achievements were made, including….

Accepted: the sentence “Using finite time thermodynamics (FTT) [1-16] to optimize the performances of practical cycles and processes has made a series of achievements” has been changed as “Using finite time thermodynamics (FTT) [1-16] to optimize the performances of practical cycles and processes, a series of achievements were made”

Comment 5: Page 5, line 116, 119 and 120:  “coincides”, instead of “are coincided”

Accepted: in line 116, 119 and 120, “are coincided” are substituted by “coincide”.

Comment 6: Page 8, last line: the more complicated the model of variable SHR, the closer the distance between model and actual cycle.

Accepted: in last line of page 8 the sentence and the more complicated model of variable SHR is, the closer the distance between cycle model and practice one is” has been changed as “and the model of variable SHR is more complicated, the distance between cycle model and practice one is more closer”

Round 2

Reviewer 2 Report

This paper deals with performance of universal reciprocating heat-engine cycle with variable specific heats ratioof working fluid. The introduction presents a good summary of previous work on the subject. After this introduction the authors describe cycle model. After this , the authors pass to the results. The authors have improved the discussions on the results. The authors have take into account all the reviewers' comments.

In my opinion this paper can be accepted in the present form for publication in Entropy.